# The Effects of Ayres Sensory Integration and Related Sensory Based Interventions in Children with Cerebral Palsy: A Scoping Review

**DOI:** 10.3390/children9040483

**Published:** 2022-04-01

**Authors:** Jiří Kantor, Lucie Hlaváčková, Jian Du, Petra Dvořáková, Zuzana Svobodová, Kristýna Karasová, Lucia Kantorová

**Affiliations:** 1Center of Evidence-Based Education & Arts Therapies: A JBI-Affiliated Group, Faculty of Education, Palacký University, 779 00 Olomouc, Czech Republic; jiri.kantor@upol.cz (J.K.); zuzana.svobodova@upol.cz (Z.S.); lucia.kantorova@mail.muni.cz (L.K.); 2Institute of Special Education Studies, Faculty of Education, Palacký University, 779 00 Olomouc, Czech Republic; lucie.hlavackova01@upol.cz (L.H.); kristyna.karasova01@upol.cz (K.K.); 3Department of Rehabilitation Medicine, General University Hospital in Prague, 128 00 Prague, Czech Republic; petra@playsi.cz; 4Faculty of Health Sciences, Palacký University, 775 15 Olomouc, Czech Republic; 5The Czech National Centre for Evidence-Based Healthcare and Knowledge Translation (Cochrane Czech Republic, Czech CEBHC: JBI Centre of Excellence, Masaryk University GRADE Centre), Institute of Biostatistics and Analyses, Faculty of Medicine, Masaryk University, Kamenice 753/5, 625 00 Brno, Czech Republic

**Keywords:** sensory integration, Ayres, sensory-based intervention, cerebral palsy, children, scoping review, movement

## Abstract

The theory of Ayres Sensory Integration^®^ was formulated in the 1960s, and is also known as sensory integration (SI). It has been used in people with cerebral palsy (CP), though the research evidence for its effects in this population is contradictory and inconclusive. To fill in this knowledge gap, we conducted a scoping review of the body of literature on the topic, including any type of quantitative or qualitative research of SI in people with CP without any restrictions of age, language, geography, professionals involved, etc. In September 2020, we searched Scopus, ProQuest Central, MEDLINE (via PubMed), CINAHL Plus and the Academic Search Ultimate and Web of Science, as well as the grey literature sources OpenGrey and MedNar. Two reviewers independently screened the texts and the references lists of the included papers. We finally included seven relevant papers (four randomized controlled trials, two quasi-experimental studies and one case series), though not all fidelity measures required for Ayres SI were reported in the papers. The age of participants ranged from 3 months to 15 years; no studies were identified on adults. There is some evidence that SI or related sensory-based interventions (SBI) may be useful for movement development and other outcomes (attention span, therapy of sensory processing disorders, body perception and therapy of strabismus), but there is only scarce and low-quality evidence comparing interventions. We recommend to conduct well-designed randomized controlled trials (RCTs) with an optimal sample size on the effectiveness of formal Ayres SI for the motor development or other outcomes (as attention span or self-care abilities) using standardized measurement tools.

## 1. Introduction

The theory of the Ayres Sensory Integration^®^ approach was formulated in the 1960s by the American occupational therapist and psychologist Anna Jean Ayres, as a therapy for patients with sensory processing disorders [1]. This approach was originally called the “sensory integrative approach” [2], but is currently known as sensory integration therapy, sensory integration treatment or simply sensory integration (SI), among other terms. The terminological inconsistency leads to frequent confusions with other types of therapy identified as “sensory-based intervention”, which is based on similar foundations or target similar manifestations of sensory processing disorders (SPD), but the intervention approach differs from that of Ayres [3,4]. Contrary to the other approaches, Ayres’ SI therapy focuses primarily on praxis and motor planning, placing a primary emphasis on the tactile, vestibular and proprioceptive system, and taking into consideration the person’s active involvement [5]. The key principles of SI parts of the Ayres Sensory Integration^®^ Fidelity Measure© encompass a total of ten rules, including, for example, physical safety, sensory capabilities and cooperation with the person on activity selection, etc. [6].

SI therapy has been used in persons with special needs, especially with autism spectrum disorders [7,8,9], but also intellectual disability and learning disorders [10,11,12], aphasia [13,14] or motor and neurological issues [12,14]. SI often targets preschool- and school-aged children. Cerebral palsy (CP) diagnosis is mostly associated with motor deficits, including impairment to motor neural pathways. However, studies show that sensory neural pathways can also be impaired, and there may be a substantial impact of frequent comorbidities such as sensory disorders, intellectual and developmental disabilities, such as epilepsy, etc. [15]. Hoon et al. even suggests that in some persons with CP, the impairment of pathways responsible for sensory perception may cause motor problems, and have a more significant impact on the functioning compared with impairment in the motor area [16]. According to Bleyenheuft et al. and Pavão et al., the most frequent deficits are in the area of tactile perception, tactile discrimination, stereognosis (identification of an object by touch) and the proprioceptive system [17,18]. Pavão et al. refer to specific deficits in sensory modulation in relation to movement and body position in space, as well as emotional reactions [18]. Other areas include behavior and registration of a stimulus, etc. Bumin & Kayihan emphasize difficulties in the right-left discrimination, proprioceptive system and praxis [19]. Kashoo & Ahmad refer to a direct correlation between the extent of attention and sensory deficits in children with hemiparesis [20]. Daly found that in various types of CP, the most significant impairment related to the vestibular and proprioceptive system with spastic forms showed more specific deficits than the ataxic and dyskinetic forms [21].

SI therapy itself is not the first choice from among interventions used in CP patients [21]. According the recent update of an overview study by Novak et al., the effective allied health interventions include bimanual training, constraint-induced movement therapy, fitness training, goal-directed training, hippotherapy, mobility training, strength training, task-specific training and treadmill training, etc. [22]. Some approaches, e.g., the Neurodevelopmental treatment (NDT), which were traditionally used for treatment of persons with CP, were found to not be effective [23]. Furthermore, SI was included into the group of non-effective treatment methods. However, there is most likely only scarce research evidence on this topic as the application of SI was for decades considered controversial in people with CP [21,24], and only recently has this situation loosened up. Some current authors, such as Daly, recommend to consider the combination of the SI approach with other interventions [21], and thus compensate for some insufficiencies of physiotherapeutic methods, including especially passive experiences [21]. Moreover, the conclusion on ineffectiveness reported in Novak et al. [22] may be influenced by a lack of included sources—the conclusion is probably mainly based on the results of a systematic review by Vargas and Camili from 1999 [14], who report only one old study [25] on children with motor delay and SI. However, we found some newer studies [20,26,27] on the topic during our preliminary search.

We also searched for any underway or completed systematic or scoping review in Epistemonikos, MEDLINE (via PubMed), Cochrane Library (Central), JBI Evidence Synthesis and Prospero. Only one review [28] from 2012 was found to have a marginal reference on the use of SI in people with CP.

In this review, we aimed to respond to this identified knowledge gap. We chose the methodology of scoping reviews to determine the extent and characteristics of the body of literature for this topic. Scoping reviews are recommended where the purpose of the review is “to identify knowledge gaps, scope a body of literature, clarify concepts or to investigate research conduct” [29]. The objective of the review is, more specifically, to explore any potential impact of SI therapy on people with CP, and to determine the type of research needed in this field.

## 2. Materials and Methods

This scoping review was prepared according to the JBI methodology [29,30], and was reported using a combination of the new Preferred Reporting Items for Systematic Reviews and Meta-analyses for systematic reviews (PRISMA) [31], as well as previous extension for scoping reviews (PRISMA-ScR) [32], described in more detail in a recent paper by Peters [33]. The development of this scoping review was based on a prospectively published protocol (available from: www.osf.io/rdz6n/, accessed on 8 November 2020).

### 2.1. Review Question

“What literature exists on the use of the sensory integration therapy in people with cerebral palsy?”

More specifically, the questions to be answered in this review are:-What are the characteristics (e.g., study design, sample size) of the existing research studies in the field?-What types of outcomes/effects (benefits and harms) of the intervention and experiences with the intervention do the studies report?-Is it appropriate to conduct a systematic review on the effectiveness and safety of the intervention? If not, what research is needed to provide more data?

### 2.2. Inclusion and Exclusion Criteria

#### 2.2.1. Participants

This scoping review included studies relating to the population of people with CP without age limitation, CP type or comorbidities. We also included studies on very young children with the diagnosis of a central coordination disorder, which often precedes the confirmation of CP. We also included research studies on a wider range of populations, providing that it was possible to distinguish between the results of persons with CP and other participant sub-groups. We excluded any studies where the purpose of treatment was not connected to the CP of participants.

#### 2.2.2. Concept

We included studies aimed at SI therapy according to Ayres, or sensory-based intervention containing elements of SI therapy according to Ayres. The therapeutic methods were not restricted due to their wide variability, but they had to focus on the stimulation of sensory systems (most frequently the vestibular, proprioceptive and tactile systems) with the person’s active involvement. We excluded studies focused only on vestibular stimulation, other (non-Ayres) sensory-based interventions and studies on assessment/evaluation without an intervention phase.

#### 2.2.3. Context

The scoping review included studies conducted in a broad geographical context and in different therapeutic settings without any restrictions on the institutional context or the primary profession of SI professionals. We assumed that SI would be practiced by therapists recruited from occupational therapists, physiotherapists, special education teachers, etc. We did not exclude any studies based on context.

#### 2.2.4. Type of Studies

The review included quantitative as well as qualitative studies, including systematic reviews. Educational and other texts, all types of non-systematic reviews and bachelor and diploma theses were excluded.

### 2.3. Search Strategy

The search was performed in November 2020 in the databases Scopus, ProQuest Central, MEDLINE (via PubMed), CINAHL Plus, Academic Search Ultimate and Web Of Science, and the grey literature sources OpenGrey and MedNar. The search strategies were designed and conducted by a professional information specialist with experience in systematic and scoping reviews. The aim of the search strategy was primarily high sensitivity to ideally capture all existing literature on the topic. Subsequently, the reference lists of all the studies included in this review were searched. No time restriction was applied. The language of the studies was not restricted, provided that the abstract was available in English. The search strategies for all sources are included in Appendix A.

### 2.4. Evidence Selection

We collected and uploaded all citations to Zotero-5.0.85, and removed any duplicates. The titles and abstracts were subsequently screened by two independent reviewers (blinded) according to the inclusion criteria defined for this review. The full texts of potentially relevant studies were retrieved and screened by two independent reviewers (blinded). The studies in Chinese were translated by a native speaker (blinded). The reasons for the exclusion of studies at the full text screening stage were recorded, and are reported in the PRISMA flow chart [31] and in Appendix A.

### 2.5. Data Extraction

The data was subsequently extracted by two reviewers (blinded) using a data extraction tool created by the authors (see Appendix A). Any doubts about the relevance of the selection and any disputes between the reviewers at each stage of the study selection and data extraction process were resolved by a discussion or by a third reviewer (blinded).

The extracted data included the title, author and year of publication of the study, geographical location and therapeutic setting, data on the number, age, gender and diagnosis of participants, research design and its JBI level, study methodology and measurement tool, the description of the intervention (by areas as specified by Shaaf and Mailoux [8]) and the findings.

### 2.6. Data Analysis and Presentation

The extracted data were presented in a narrative and tabular form in a manner consistent with the objectives of the scoping review.

## 3. Results

In the development of this scoping review, we identified a total of 2572 texts from databases, 411 grey literature papers and 2 texts from reference lists of included studies (the number of records for each source is specified in Appendix A). After the removal of 1000 duplicates, the number of records for screening was 1843. 18 full texts were retrieved, of which 11 were excluded, with reasons reported in the PRISMA flow chart below (Figure 1). The scoping review included seven studies.

The studies were published between 2000–2020. Although labelled as Ayres SI, none of the studies reported full adherence to the rules of Fidelity Measure©, according to Ayres (we will therefore use the term SI/SBI to report the relevant studies in the next sections). Two studies were from Iran, one was from Turkey, one from Saudi Arabia and three from China. All of the studies included child participants (aged 3 months to 15 years) diagnosed with one of the forms of CP. The number of participants ranged from 17 to 81. As far as the number of participants is concerned, the studies carried out in China [27,34,35] included a higher number compared with the other studies. However, these studies focused on persons with CP across the different types of this impairment, but none of the types was prevalent. The other studies [19,20,26,36] each focused on one type of CP. There were 66 participants with diplegia, 11 with quadriplegia and 17 with hemiplegia. All participants were children, in most cases up to 7 years; one study [20] focused on participants aged 10 to 15 years. This study focused on the effect of SI therapy on attention span in persons with CP, which is an ability frequently examined in older children in the context of school attendance. As for comorbidities, in the study by Zhou et al. [35], a significant comorbidity was strabismus. Comorbidities in other studies were not mentioned. All of the studies included both girls and boys.

In terms of design, there were three randomized controlled trials (RCT) [26,27,36], one controlled trial with inadequate randomization [19], two quasi-experimental studies [20,34] and one case series [35]. We did not identify any qualitative studies. Most studies, including the trials, contained severe methodological limitations. The randomization method was not specified in any of the RCTs. Furthermore, two of them did not contain any further information about blinding and allocation assignment [26,36]. The study of Bumin & Kayihan [19] used systematic assignment into groups according to the date of admittance to the clinic, so therefore we describe it as a pseudo-randomized controlled trial.

There was heterogeneity in terms of the control interventions. One study used NDT [36], another a combination of NDT and the Vojta method [27], and another study conventional physical therapy [20], which consisted of gait training, exercise sessions on Swiss Ball, stretching and Mat exercises. The control group in the study by Shamsoddini et al. [26] completed a routine occupational therapy home programme, including activities to maintain a position resting on the forearms and hands in the sitting position, a semi-kneeling position, during crawling and in the standing position. These positions were assisted by an occupational therapist or parent until the muscle tone was reduced. Balance and position change response were developed through activities on a gymnastic ball and balance board. In the studies by Chen et al. [34] and Bumin & Kayihan [19], one group took individual SI therapy while the other was involved in group SI/SBI therapy. The study by Bumin & Kayihan [19] had a third group that underwent a home-based programme with selected activities applied in the intervention groups. The case series by Zhou et al. [35] included two age groups (8–18 months and 19–36 months), where both received SI therapy (with no control group).

The duration of one SI/SBI session was between 30 min [27] and 1.5 h [19,36]. Other studies applied sessions of 60 min [20,26,35] and 40 min [34]. The highest frequency of therapy was 20 sessions per week in the study by Zhou et al. [35], and the lowest frequency of therapy occurring in most of the studies was 3 sessions per week [19,20,36]. The most frequent length of the treatment was 3 months [19,26,34,36]. The longest therapy duration was 5 months [35].

The authors used different kinds of outcome measures. Gross motor skills were measured by the Gross Motor Function Measure (GMFM) [26,27,36]. The ability of self-care was measured by the Physical Ability Test (PAT), which allows the assessment of activities of daily living according to age groups [19]. Different body functions and sensory integration problems were measured by the Ayres Southern California Sensory Integration Test (SCSIT) [19], the Berg Balance Scale (BBS), which measures static and dynamic balance on a 14-item performance observation measure [34], and the Arabic version of the Stroop Color and Word Test, which measures the attention span [20]. The overall development in four areas of Gesell Developmental Schedules translated into Chinese was a measured study by Zhao et al. [27], and Zhou et al. [35] used the strabismus curative effect evaluation standard (confirmed by Chinese Academy of Ophthalmology). Because the children with CP aged under 3 years old did not cooperate, the authors used the method of corneal reflection and occlusion to measure the strabismus of 33 cm and 6 m (recorded by digital camera) [35].

All the measurements were applied pre- and post-treatment. The follow-up measurement was only done by Kashoo & Ahmad [20] and Zhou et al. [35]. Kashoo & Ahmad [20] mentions the 4-month follow-up in the abstract, and Zhou et al. [35] one year in the main text, but there is no further information in either of the papers.

### 3.1. Types of Sensory Integration Interventions

The studies included mostly individual form of SI/SBI, but two groups also applied a group form of therapy. Interventions focused on sensory perception (with the aim to enhance tactile, proprioceptive and vestibular perception), postural and ocular control and bilateral integration, as well as praxis. Examples of activities included:Tactile perception—e.g., experience with objects of different shapes, sizes and surfaces, tactile walkways and recognition of friends with eyes blindfolded.Proprioceptive perception—e.g., tracing of the body, “window game”, pressure on body parts and identification of weight of known objects.Vestibular perception—e.g., activities on a gymnastic ball, balance board, swing, trampoline, climbing wall, rocking horse, practicing balance in various positions, standing on one leg, walking in sand or water or walking backwards.Postural and ocular control and bilateral integration: e.g., turning to the left and right, ball games, hitting a ball into a basket and target, training of head, neck and chest control in the central position.Praxis: e.g., ball games, for example throwing a ball with a leg, wheel barrow, swimming/drying, inchworm art, stick ball, stringing beads, touch boards, writing in different positions, tear art in the kneeling position, doing up buttons, tying knots, tracing and finger drawing.

The intervention in the study by Zhao et al. [27] differed by participants’ age. Participants aged 0–6 months had second-level training—integration of tactile, vestibular and proprioceptive perception, while participants aged 7–12 months received third level training—integration of the visual system, hearing system and the three systems mentioned above. In Kashoo & Ahmad [20], each activity was preceded by ten-minute group stretching. In group SI in the study of Chen et al. [34], the activities were adapted to a game with rules involving three-member groups. The intervention approach was selected according to the type of the sensory processing disorder that participants were diagnosed with (classification by Miller et al. [1]), which makes the indication of a specific intervention more individualized. In the study by Shamsoddini [36], the intervention was not described, and the study by Zhou et al. [35] described the intervention as merely a “standardized SI therapy”. All activities are described in detail in the data extraction tables in Appendix A.

### 3.2. Findings Reported in the Included Studies

The findings of all studies were divided into two groups: outcomes connected to gross motor skills; and other outcomes. Three studies reported outcomes connected to the gross motor skills according to the domains of GMFM, and one study also included original reflex (asymmetrical tonic neck reflex, symmetrical tonic neck reflex, tonic labyrinthine reflex, Moro), balance reflex and postural reflex (parachute reflex) [27]. All studies reported some improvement of gross motor skills in the SI/SBI group (Table 1). The study of Zhao et al. [27] also says that the results depend on the cooperation with parents, and it is crucial to enhance the medical and educational training for them.

Other studies reported findings on everyday activities, sensory integration and balance, perception of one’s body, communication abilities and development, attention and strabismus therapy. All studies reported some improvement after SI/SBI intervention, and some also reported statistically significant improvement when compared with the control group (Table 2).

Results of the study by Bumin & Kayihan [19] confirms that CP is connected to visual and somato-sensory perception dysfunction—participants in all three groups obtained low scores from design copying, position in space, graphesthesia, tactile stimuli perception, manual form perception and finger identification (which are used to assess the sensory perception). Furthermore, in a study of Chen et al. [34], participants were divided into smaller groups (3 children in a group) according to the children’s sensory integration disorder classification (sensory modulation disorder, sensory discrimination disorder or sensory-based motor disorder). According to the same study [34], having the children discuss how to design the activities leads to them rehearsing the movements in their minds, and an improvement in their ability to plan movement. In such an atmosphere of help, cooperation and competition, each child can fully develop their own potential, shape good character and improve social communication skills.

Figure 2 offers a summary of all the outcomes reported in the studies. According to the International Classification of Functioning, Disability and Health [37] the outcomes are represented by two main domains: functions and activities. Functions are represented mainly by sensory functions (e.g., seeing, tactile perception, or proprioceptive functions), mental functions (e.g., attentions span) and movement functions (motor reflex functions). Activities are represented by mobility (changing basic body position, walking and moving around). Some outcomes belong to more complex categories such as self-care, language development, social-emotional responses, etc.

## 4. Discussion

Seven studies were eligible for this scoping review. All studies focused on children with CP (302 children counted in all studies, no studies were found for adults), and applied an individual as well as a group form of SI/SBI. Standard length of the therapy was 3 months, and the length of the treatment was 30–90 min. There were three RCTs, one pseudo-RCT, two quasi-experimental trials and one case series, and no qualitative studies were identified. Only Kashoo & Ahmad [20] reported limits of their study, though several other studies contained obvious methodological problems. Formal quality assessment was not done in this review. There was heterogeneity of interventions, but a specific focus on individual and group form of SI/SBI was found in some studies. Concerning the outcomes, the studies reported on gross motor skills, everyday activities, deficits in sensory integration, balance, communication skills and child´s development, attention span and strabismus. There was an expected focus on different types of functions and activities (mainly mobility). Whereas the mobility was assessed only by GMFM, there was a heterogeneity of tests used for different body functions, e.g., by PAT, WDI, SCSIT, Stroop Color and Word Test or Gesell Developmental Schedules. The included studies used various types of comparators (e.g., NDT), and their results show different trends. No study was reported on safety/adverse events, and qualitative studies are completely missing.

Three studies [26,27,36] found a positive impact of SI/SBI on motor development and movement skills. There may be questions regarding what kinds of SI/SBI intervention could have a positive impact on movement. We thought this could be explained by the influence of vestibular stimulation techniques that were included in SI/SBI interventions in all above-mentioned studies. The effects of vestibular stimulation on persons with CP have been reported by a number of systematic reviews (such as Topley et al., 2020 [38]; and Ankna et al., 2018 [39]). However, one of the studies [26] included the vestibular stimulation techniques in both the intervention as well as the control group, and still the results of this study indicated an increased improvement in the SI/SBI group compared to the control group. If this is true, then other SI/SBI interventions could also have an impact on movement ability.

Two studies [19,34] compared the effects of individual versus group form of SI/SBI. While the study by Bumin & Kayihan [19] observed only small differences in the results of both groups, the study by Chen et al. [34] reported a significant difference in favor of the group form of intervention. So far, SI/SBI is mostly conducted on an individual basis, so this finding could be crucial for the future development of this approach. However, further research and verification is needed to test the credibility of these conclusions.

Findings of Kashoo & Ahmad [20] suggest there is some potential of SI/SBI to influence attention span. A low level of attention may be a manifestation of sensory process disorders, as reported by Louwrens [21] in a study on children with hemiparesis. Although attention is not a typical outcome in the research of SI/SBI, there is some evidence concerning the effect of SI/SBI on cognitive functions in children with autism spectrum disorders and other developmental disabilities [11,40].

Some interesting conclusions were suggested by further extracted data. As far as the period of publication is concerned, the studies included in the scoping review can be divided into two categories. Five studies were published at the beginning of the 21st century [19,26,27,35,36], while two studies are more recent [20,34]. Currently, this theme is moving to the center of attention. This fact is confirmed, for example, by the study by Louwrens [41], which described sensory process disorders (SPD) in children with CP, and advocated the applicability of SI/SBI therapy in this target group. The necessity to focus therapy on sensory processing is also reported by other studies focusing primarily on the types and manifestations of SPD in persons with CP [17,18]. No study on adults with CP was found, though various complications associated with SPD also manifest in the adult age, but there is very limited evidence [42,43,44].

SI/SBI therapy is approached in different ways across the world. For example, in the US the rules for this intervention are stricter, and as has already been mentioned, Ayres SI therapy is usually not typically associated with CP diagnosis. There is a clear distinction between the Ayres sensory integration therapy and the so-called sensory-based intervention [3] that, according to Ayres, differs from SI [5]. None of the studies found in this review reported full adherence to the rules of Fidelity Measure©, according to Ayres. Therefore, it was not possible to validate if the studies were done in compliance with the basic principles of Ayres SI.

In this scoping review, we found some evidence about the positive effects of SI/SBI (mainly in the study of Zhao et al. [27]), but there is still inconclusive evidence if SI/SBI has any benefit over control interventions. Overall, the studies were very heterogenous in terms of design, age groups, number of participants, specific interventions and comparisons, and the measured outcomes, measurement instruments and tools were of low quality (although not formally appraised in this review), and poor statistical methods limited the interpretation of the results. More research is needed to determine the true effects of SI/SBI.

### 4.1. Implications for Research

In future studies, it is necessary to examine the effect of SI/SBI compared to usual treatment on motor development in children with CP. We suggest to conduct an experimental or at least quasi-experimental study with optimal information size in children with spastic CP. The optimal information size is according to the GRADE methodology, increasing the precision of study results, and so increasing the overall certainty of evidence. The newly designed study should reach the highest possible certainty defined by GRADE to prevent the research waste [45,46]. In the case of RCT, at least data measurement should be blinded, and the allocation concealment performed. SI/SBI should optimally adhere to the key principles of the Ayres Sensory Integration^®^ Fidelity Measure© [6], and should be performed by an Ayres Sensory Integration-certified therapist. The outcomes should be measured by GMFM, as has been the practice in this area until now. We recommend not using NDT or similar approaches as a control intervention, as it was found not as effective by Novak et al. If conventional physiotherapy is used as a control, it should be based on the active participation of the child during the therapeutic process.

It could thus be useful to research the potential of SI/SBI not only in terms of motor development, but also cognitive abilities (attention span), sensory processing disorders, communication and overall development, safety and possible adverse effects.

Based on the experiences from published studies, the lowest frequency of therapy should not be less than 3 sessions per week, the sessions shouldn´t be shorter than 30 minutes, and the length of the therapy should not be shorter than 3 months. Although individual form of SI/SBI is mostly offered in practice, the researchers may also consider the group form of therapy. There is inconclusive evidence that some form of therapy should be superior, and the difference in forms of treatment should be researched in future studies. Similarly, there is not enough data to suggest specific interventions that would impact different types of outcomes.

It is necessary to improve the methodological quality of the studies; future authors should perform the statistical comparison between experimental and control groups, not only pre-test and post-test analysis. Future studies should apply current standardized outcome measures, e.g., Goal Attainment Scale [47]. The GMFM scale could be enriched by activities of daily living measurement and quality of life that is important for SI/SBI, as well as for occupational therapy. We do not recommend developing a systematic review of effectiveness until there is more evidence. If SI/SBI proves to be effective enough for people with CP, the research of various SI/SBI forms and interventions would be needed.

### 4.2. Strengths and Limitations of the Scoping Review

We used the Preferred Reporting Items for Systematic Reviews and Meta-analyses extension for scoping reviews (PRISMA-ScR) [32] for reporting (Appendix A). The strengths of the scoping review include compliance with the standard JBI methodology for scoping reviews [30]. The review conducted a robust search by a professional information specialist, and included grey and unpublished literature that was sensitive and inclusive. The team of authors was composed of specialists in occupational therapy, special education, medicine, psychology and evidence synthesis methodology, as well as an experienced practitioner in Ayres sensory integration. The studies published in Chinese were translated by a native speaker.

The results of this review are limited by a small number of studies found, notwithstanding a sensitive search strategy and a considerable number of information sources searched. Other limitations of the scoping review include the fact that the authors of the studies included were not contacted to complete the missing information. We did not have enough information connecting the adherence of the SI therapy to the requirements of the Ayres Sensory Integration^®^ Fidelity Measure© [6]. No included study specified whether the therapy had been conducted by an Ayres Sensory Integration-certified therapist, and it was impossible to assess compliance with the principles, including, for example, a just-right challenge, active child involvement in the intervention, establishment of therapeutic alliance, challenge to praxis and organization of behavior of the child, etc. which are crucial to the results of the intervention [8]. Moreover, the studies don´t contain any important information about data analysis or the statistical comparison of results between experimental and control group.

## 5. Conclusions

The results of the scoping review show that SI/SBI has a potential for rehabilitation for people with CP (mainly in the area of motor skills), but there is only scarce and low-quality evidence comparing interventions. Future research on the effectiveness and safety of SI/SBI in people with CP is needed. We recommend to conduct an experimental or quasi-experimental study in children with spastic CP concerning the motor development or other outcomes as attention span or self-care abilities. There is a need to report if the SI/SBI provided adhered to the criteria of the Ayres Sensory Integration^®^ Fidelity Measure©, and to improve the methodological quality of the research. We do not recommend conducting a systematic review until there is more well-conducted primary research.

## Figures and Tables

**Figure 1 children-09-00483-f001:**
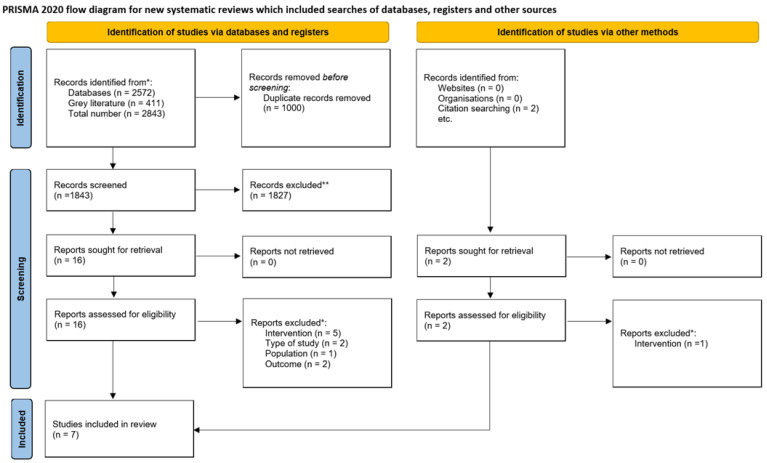
PRISMA flow chart (*, ** for a list of excluded studies and the reasons for exclusion, see Appendix A).

**Figure 2 children-09-00483-f002:**
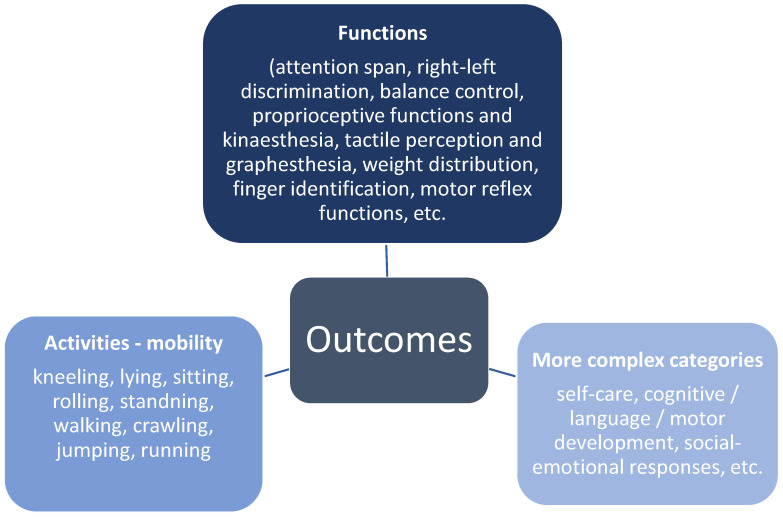
Summary of outcomes.

**Table 1 children-09-00483-t001:** Outcomes connected to the gross motor skills after SI/SBI. Abbreviations: CG: Control group; GMFM: Gross Motor Function Measure; NDT: Neurodevelopmental treatment.

Author	Outcome Measure	Control	Results
Shams and Holisaz [26]	GMFM	physical exercise	The overall score: The improvement after treatment: SI/SBI group 22.5 points, CG 1.9 points.
			The domains of GMFM: Significant between groups difference favoring the SI/SBI group in: sitting (*p* = 0.02), crawling (*p* = 0.001) standing (*p* = 0.03). Pre-test/post-test difference—SI/SBI group had a statistically significant improvement in: rolling (*p* = 0.001), sitting (*p* = 0.04), crawling (*p* = 0.002), standing (*p* = 0.01). CG only showed a statistically significant improvement in rolling (*p* = 0.001).
Zhao et al. [27]	GMFM original reflex balance reflex postural reflex	A combination of NDT and Vojta method (experimental group received received SI/SBI and NDT).	Between groups difference: Significant difference in overall score (*p* < 0.001) favoring the SI/SBI group, namely in: lying, sitting, crawling, kneeling position. Significant difference in the original reflex, balance reflex and postural reflex.
Shamsoddini [36]	GMFM	NDT	Pre-test/post-test difference: improvements in all domains of GMFM in both groups. Between groups difference: Only in the domain of standing seems to be significant difference (14 points favoring NDT group). The statistics contained obvious inconsistencies and the validity of data is low.

**Table 2 children-09-00483-t002:** Other outcomes after SI/SBI. Abbreviations: BBS: Berg Balance Scale; N/A: not applicable; PAT: Physical Ability Test; SCSIT: Ayres Southern California Sensory Integration Test; WDI: weight distribution index. Ayres Southern California Sensory Integration Test.

Author	Outcome Measure	Control	Results
Bumin & Kayihan [19]	SCSIT PAT	3 groups: individual SI/SBI, group SI/SBI and home programme	Between groups difference: Groups applying SI/SBI improved more than the control group, but there was a low difference between the group form and individual form of SI/SBI. Pre-test post-test difference in individual and group SI/SBI: Improvements in most SCSIT, namely in double tactile stimuli perception, graphesthesia, kinaesthesia, finger identification, design copying, position in space, imitation of posture, right-left discrimination. Statistically significant deterioration: localization of tactile stimuli and motor accuracy. For manual form perception, all three groups showed improvement without statistical significance. The PAT scores showed a statistically significant improvement for all three groups. Although the authors of the study offer many statistical values (mean differences, confidence intervals, *p* values, standard deviation, standardized mean differences), the statistical comparison among the three groups is missing, and the inter-group differences may only be evaluated on the basis of descriptive comparisons.
Chen et al. [34]	BBS WDI	Individual and group SI/SBI without any control group	Between groups difference: Group form of therapy was found more beneficial than the individual form. Pre-test post-test differences: A statistically significant difference found in: BBS scores in both groups (*p* < 0.001). Two WDI categories—HL (head turned left, *p* < 0.001) and HB (eyes closed, head tilted back, *p* = 0.002). Another two WDI categories were close to significant difference—NO (basic position with eyes open, *p* = 0.074) and NC (basic position with eyes closed, *p* = 0.073). Only the HF category (eyes closed, head bent forward, *p* = 0.434) was far from statistical significance.
Zhao et al. [27]	the Gesell Developmental Schedules	NDT and Vojta therapy	Between groups difference: Significant improvement (*p* < 0.05) in SI/SBI group in four categories of the Gesell Developmental Schedules: respondence human ability; respondence object ability; action ability; language ability.
Kashoo & Ahmad [20]	Stroop Color and Word Test	conventional physical therapy	Between groups difference: Statistically significant differences in all items of the Stroop Color and Word Test favoring SI/SBI group: correct answers (*p* < 0.001) incorrect answers (*p* < 0.0001), no answers (*p* < 0.0001), reaction time of matched words (*p* < 0.001), and reaction time of unmatched words (*p* < 0.0001). Change in the scores was retained after 4 months follow-up.
Zhou et al. [35]	Measurement of corneal reflection and occlusion	N/A Only comparisons of two groups divided according to children’s age	15 eyes (13%) were functionally cured, 83 eyes (72.2%) were partially functional cured, 17 eyes (14.8%) were ineffective, and the total effectivity rate was 85.2%. There was a significant difference between infant group (8–18 months) and young children group (19–36 months), (*p* < 0.005) (younger children improved more). There was no significant difference between strabismus type and curative effect (*p* > 0.05).

## Data Availability

Not applicable.

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
