# Peer review of "The Effects of Ayres Sensory Integration and Related Sensory Based Interventions in Children with Cerebral Palsy: A Scoping Review"

_children, 2022, doi:10.3390/children9040483_

Round 1

Reviewer 1 Report

I appreciate the considerable effort authors have made to review literature on use of sensory-based interventions in children with cerebral palsy. Unfortunately, methodology used is inappropriate to answer the question posed, which critically weakens this study's ability to draw conclusions. In particular, the study's aim to examine effects of intervention is a quantitative question-- and as such requires systematic (rather than scoping) review to answer. JBI recommendations as cited by authors highlight types of questions that can be effectively answered using scoping reviews; for example, a narrow focus on tabulating the types of outcomes that have been examined when studying sensory interventions and/or the types of interventions that have been studied. As a result of this mismatch, authors' conclusions are overstated and not supported by their analysis (e.g. their recommendation to use SI/SBI as part of treatment)

Additional comments by section:

Introduction:

-Review of current evidence for therapies used in children with CP is out of date. See for example systematic review by Novak et al., 2020, Pediatric Neurology; NDT, for example, is considered not to be effective. Note also, however, that sensory integration therapies were found not to be effective.

Methods:

-Processes regarding screening/ultimate inclusion of studies raise substantial questions. For example, as inclusion criteria did not restrict inclusion to Asian studies, it is unclear why all studies ultimately included in this analysis were performed in Asia. Inclusion of 7 studies out of 1945 non-duplicate records screened seems to indicate over-inclusive search methodology as well as a restrictive eligibility process that may have missed a substantial number of relevant studies.

Results:

-Mismatch between question asked and methods used becomes apparent; in 3.2, authors largely only summarize results. Types of results as reported in 3.1 are more appropriate for a scoping review.

Discussion:

-Authors laudibly note methodological lapses contained within included studies. However, this should extend to further limit the strength of any conclusions reached. For example, if outcomes are of "low quality" and statistical analyses are "poor", how can it be suggested that studies  show   "potential superiority" to other forms of physiotherapy?

Author Response

Dear Sir or Madam,

Please see the attachment. Thank you so much.

Best

Reviewer 2 Report

This paper presents an original literature review, conducted according to JBI Methodology, on the usage of Ayres sensory integration (SI) for people with cerebral palsy (CP) – a population that has mainly been treated exclusively by the Neurodevelopmental treatment (NDT). The authors intended to explore whether SI therapy has a sufficient benefit for people with CP to justify further research of SI effects and SI effectiveness compared to other forms of therapy. In this review, the authors found only seven papers (out of 1,945 results), searching in six databases and two grey literature sources. According to these authors, there is some evidence that SI may be useful for movement development and other outcomes such as attention span, therapy of sensory processing disorders, body perception, and therapy of strabismus. The paper is well structured and easy to read. In addition, it presents in detail and adequately discusses the results found in the reviewed literature. As a weak point, I point out the small number of papers found in this literature review that limits its contribution to the area of ​​knowledge.

Author Response

(The authors gave the same response as above.)

Reviewer 3 Report

Thank you for inviting me to review this study, which I found very interesting as it is a tool that is widely used in clinical practice, but where the scientific evidence is still not clear. But I have doubts when assessing the manuscript.

Abstract: these terms should not appear in this section "in favor of Neurodevelopmental treatment".

Introduction: it would be interesting to add references on the co-morbidities of sensory disorders. Novak’s studies are not referenced; which discourage sensory integration; and even in this study, NDT is mentioned, when in Novak et al. is also not recommended NDT. It does not make much sense to say that there are studies that support its use, when there is only one systematic review that has assessed it, therefore, many statements cannot be made.

Methodology: there is no mention of exclusion criteria at the beginning of this section, but later it is commented that there have been. It is contradictory.

Results: the flowchart is not the current one, it would have to be changed. Define pseudo-RCT. It would be interesting at the beginning of this section to say the total number of subjects by topography (using the current terms of unilateral and bilateral), in addition to the total number of children. The paragraph where Zhou et al. is mentioned should be omitted and certain aspects should be removed as there is no control group as such. The paragraph of the tools to assess, could be made based on ICF or by common aspects to assess. Grouping the results by outcome would be very interesting; so we would have to rewrite this entire section. The table should be more concise and clearer; in addition to appearing less text and more data.

Discussion: results are given that should not be in this section, and if in result. The quality of the studies is discussed, but it has not been previously analyzed or reflected in the text. It should be rewritten based on the interventions. In limitations, we talk about valuing in GMFM, and perhaps it would also be interesting to measure participation. It is very risky to talk about a lot of potential with so little evidence

Supplementary material: The search table, if compared to the diagram, does not come out the same

Regards

Author Response

(The authors gave the same response as above.)

Round 2

Reviewer 1 Report

This manuscript is substantially improved:

-Aims as stated in 2.1 are now reasonable for a scoping review;

-Methods description is substantially improved;

-Results are adequately described;

-Conclusions now largely flow from Results but perhaps can be refined somewhat

I have only minor recommendations:

-General review for clarity, organization, and English language usage. Language at times is informal (e.g. use of contractions), and at times, meaning is unclear (e.g., what is "optimal information size" in Line 611?)

-This claim is vague and may be overstated given the weakness of constituent studies:"All studies consistently showed promise in using SI / SBI"

-I agree with Conclusions that "there is only scarce and low quality evidence comparing interventions."-- the Abstract should similarly reflect this conclusion.

-Authors note substantial heterogeneity in forms of SI/SBI studied. Will this continue to be an ongoing challenge to assessing evidence for/against benefit from SI/SBI? Do authors have suggestions regarding which form(s) of SI/SBI merit further study, or, at least, how treatment factors (e.g. treatment form, setting, dosage) should be standardized and reported in order to support generalizability? Could specific forms of SI/SBI be expected to impact differing types of outcome measures (e.g. motor vs. cognitive as Authors suggest)?

Author Response

Dear Sir or Madam,

Best regards,

Jian

Reviewer 3 Report

Thank you for the reconstruction of the work, now I think it is much better and updated to the terms and conditions of the current scientific evidence.

I think that in section "3.1. Types of sensory integration interventions", the "etc." should not be reflected; and if "for example". Since "etc." it gives rise to think that not all the information of the articles has been read well

Regards

Author Response

Dear Sir or Madam,

Thank you for the suggestion and help with the improvement of our paper. Based on your feedback, we have revised it. Please check it.

Sincerely yours,

Jian